**DOI: 10.1038/ncomms15766**　　**OPEN**

# New Martian valley network volume estimate consistent with ancient ocean and warm and wet climate

Wei Luo[1], Xuezhi Cang[1] & Alan D. Howard[2]

The volume of Martian valley network (VN) cavity and the amount of water needed to create the cavity by erosion are of significant importance for understanding the early Martian climate, the style and rate of hydrologic cycling, and the possibility of an ancient ocean. However, previous attempts at estimating these two quantities were based on selected valleys or at local sites using crude estimates of VN length, width and depth. Here we employed an innovative progressive black top hat transformation method to estimate them on a global scale based on the depth of each valley pixel. The conservative estimate of the minimum global VN volume is $1.74 \times 10^{14}\,m^3$ and minimum cumulative volume of water required is $6.86 \times 10^{17}\,m^3$ (or $\sim 5\,km$ of global equivalent layer, GEL). Both are much larger than previous estimates and are consistent with an early warm and wet climate with active hydrologic cycling involving an ocean.

[1] Department of Geography, Northern Illinois University, Davis Hall 120, DeKalb, Illinois 60115, USA. [2] Department of Environmental Sciences, University of Virginia, Charlottesville, Virginia 22904, USA. Correspondence and requests for materials should be addressed to W.L. (email: wluo@niu.edu).

Fluvial landforms such as valley networks (VNs), outflow channels and delta deposits found on Mars offer the best evidence for its past water activities[1–3]. The inventory of water on Mars has been estimated based on a number of different sources, including volatiles from volcanic activities[4], geomorphologic traces left by past water activities (from paleolakes and deltas[5,6], outflow channels[7,8] and VNs[9]), the hypothesized northern ocean[10] and, more recently, the observed deuterium/hydrogen (D/H) enrichment[11]. VNs on Mars are river-valley-like features that distribute predominantly on the ancient cratered southern highlands and have been long recognized as the best evidence for past water activities[7]. In the scenario of a warm and wet early Mars climate, the water running through the VNs would eventually flow towards and collect at the topographically low northern plains, forming an ocean covering nearly 1/3 of the surface[12,13]. However, the ocean hypothesis has also been controversial[3]. Supporting evidences include the shoreline features identified from remote-sensing images[12], delta deposits found at similar elevation[14], sediment stratigraphy consistent with distributary environment[15], spatial distribution of VN termini near proposed shorelines[16] and elemental distribution of K, Th and Fe revealed by Gamma Ray Spectrometer that are consistent with paleo-ocean boundaries[17]. Evidence against the ocean hypothesis include variations in the identified shoreline elevations of up to a couple of kilometres[10,18], low water inventory[19] in comparison with the purported ocean volume and large boulders observed on the ocean floor[20]. The variations in shoreline elevation have been recently explained by later deformation associated with true polar wander[21] and tsunami deposits[22]. The presence of large boulders in the ocean floor could be emplaced by catastrophic mass-transport events similar to those documented within continental margins on Earth[23].

The volume of VN cavity and the amount of water needed to create the cavity by erosion are of significant importance for understanding the early Martian climate, the style and rate of hydrologic cycling and the possibility of an ancient ocean[10,12,14,24]. However, these two quantities on a global scale remain less well constrained. Improving these estimates may provide additional evidence or constraints regarding early Martian climate and the ocean hypothesis. The volume of VNs is most critical, because it is the basis for inferring the volume of water needed to create the VNs. Early attempts at estimating VN volume were primarily conducted at selected local sites based on simple estimates of VN lengths, widths and depths by assuming VN wall slopes[9,25,26] or based on the valley area and the average depth[27]. The results contained large uncertainty due to crude methodology and/or poor data quality. Rosenberg and Head[24] recently estimated the cumulative volume of water needed to carve the late Noachian VNs based on a fluid/sediment flux ratio function derived from terrestrial empirical data. Their estimate of

the most probable cumulative water volume was 3–100 m global equivalent layer (GEL) and concluded that the Late Noachian Martian climate may have been less wet than previously thought[24]. However, their volume of VN excavation was based on eight largest VNs analysed in a previous study[28], not on all the VNs mapped globally, even though they claimed that the rest of the VNs were small and had negligible contribution to the total global volume. Here we employed an innovative progressive black top hat (PBTH) transformation method to estimate the depth of each valley pixel[29], the minimum volume of material that would have been excavated to form the global VNs[16,30] and the minimum cumulative volume of water required to do that.

## Results

**VN cavity volume.** The global VN cavity volume estimate based on the topographically derived version of VN[16] is $(1.74 \pm 0.8) \times 10^{14}$ m$^3$ and that based on the VN that integrates both the topographically derived[14] and manually digitized[30] VN is $(2.23 \pm 1.0) \times 10^{14}$ m$^3$ (Table 1). Both estimates are one order of magnitude larger than that used the Rosenberg and Head study[24]. The errors in Table 1 were estimated based on propagation of the vertical error ($\sim 45$ m) estimated from gridded Mars Orbiter Laser Altimeter (MOLA) digital elevation model (DEM)[31]. As the horizontal error of MOLA data ($\sim 100$ m) is less than the cell size, it would not impact the final volume estimate.

**Minimum cumulative volume of water needed to carve the VNs.** There are several ways to convert the volume of VNs to the minimum cumulative volume of water needed to carve the VNs, for example, using a simple water-to-sediment ratio[9,26] or fluid/sediment flux ratio function empirically derived based on terrestrial data[24]. As we are only interested in the global scale estimate, we derived the minimum cumulative volume of water by assuming a reasonable sediment load and density of sediment[27], and the result is $\sim 5$ km GEL (based topographically derived VN) to $\sim 6$ km GEL (based on combined VN, Table 1). The minimum cumulative volume of water needed to erode the VNs is about 4,000 times the volume of VNs (for both the topography-based VN and the combined VN), suggesting a relative high rate of water recycling involved in excavating the Martian VNs and consistent with a large open water body (ocean).

If we plug in our most conservative VN volume estimate data $(1.74 \times 10^{17}$ m$^3)$ into the empirically fitted fluid/sediment flux ratio function of Rosenberg and Head (their equation (4))[24], we would also obtain larger cumulative water volume estimates, ranging from 0.6 km GEL ($\alpha = 60$), 1 km GEL ($\alpha = 35$), to 11 km GEL ($\alpha = 6$), depending on the value of $\alpha$. The parameter $\alpha$ is a

**Table 1 | Global Martian VN volume estimates based on PBTH method and volume of water required.**

| | Topographically derived VN[16] | | Combined VN (topographically derived[14] and manually digitized[30]) | |
|---|---|---|---|---|
| | Volume or mass | GEL (m)* | Volume or mass | GEL (m)* |
| $V_{VN}$ (MOLA) | $(1.74 \pm 0.8) \times 10^{14}$ m$^3$ | 1.20 | $(2.23 \pm 1.0) \times 10^{14}$ m$^3$ | 1.54 |
| $V_{VN}$ (HRSC$^\dagger$) | $(2.31 \pm 1.1) \times 10^{14}$ m$^3$ | 1.59 | $(2.96 \pm 1.4) \times 10^{14}$ m$^3$ | 2.04 |
| $V_s = V_{VN}/(1 - \lambda)$ | $(3.55 \pm 1.6) \times 10^{14}$ m$^3$ | 2.45 | $(4.55 \pm 2.1) \times 10^{14}$ m$^3$ | 3.14 |
| $M_s = V_s \times \rho_s$ | $(1.03 \pm 0.5) \times 10^{18}$ kg | — | $(1.32 \pm 0.6) \times 10^{18}$ kg | — |
| $V_w = M_s/L_s$ | $(6.86 \pm 3.2) \times 10^{17}$ m$^3$ | $4.74 \times 10^3$ | $(8.80 \pm 4.1) \times 10^{17}$ m$^3$ | $6.08 \times 10^3$ |

GEL, global equivalent layer; HRSC, High/Super Resolution Stereo Colour Imager; $L_s$, sediment load in water $= 1.5$ kg m$^{-3}$; MOLA, Mars Orbiter Laser Altimeter; $M_s$, mass of sediment; PBTH, progressive black top hat; VN, valley network; $V_s$, volume of sediment; $V_{VN}$, volume of VN; $V_w$, volume of water; $\lambda$, porosity $= 0.35$; $\rho_s$, density of sediment $= 2,900$ kg m$^{-3}$.
*Errors for GEL are all less than $10^{-6}$ m.
$\dagger$Scaled based on regression line shown in Fig. 3c.

**Table 2 | Comparison of VN volume estimates with previous studies.**

| VN location | Volume1 (m³) (Hoke et al.[28]) | Volume2 (m³) (this paper) | Vol2/Vol1 | Volume3 (m³) (Matsubara et al.[32]) | Vol2/Vol3 |
|---|---|---|---|---|---|
| 12°S, 12°E (Evros) | $9.6 \times 10^{12}$ | $1.48 \times 10^{12}$ | 0.15 | $3.35 \times 10^{12}$ | 0.44 |
| 7°S, 3°E | $7.2 \times 10^{11}$ | $1.38 \times 10^{12}$ | 1.91 | — | — |
| 3°S, 5°E | $2.8 \times 10^{11}$ | $4.16 \times 10^{11}$ | 1.49 | — | — |
| 0°N, 23°E | $1.5 \times 10^{12}$ | $7.19 \times 10^{11}$ | 0.48 | — | — |
| 2°N, 34°E (Naktong) | $8.5 \times 10^{12}$ | $3.01 \times 10^{12}$ | 0.36 | — | — |
| 12°N, 43°E | $1.7 \times 10^{12}$ | $7.88 \times 10^{11}$ | 0.46 | — | — |
| 6°S, 45°E | $2.1 \times 10^{12}$ | $7.49 \times 10^{11}$ | 0.36 | — | — |
| 24.9°S 343.7°E (SPL) | — | $6.36 \times 10^{12}$ | | $1.39 \times 10^{13}$ | 0.46 |

SPL, Samara, Parana, Loire Valles summed; VN, valley network.
Volume1 was from Table 3 of Hoke et al.[28], Naktong east and Naktong west were summed here; volume2 was this study based on the combined VN; volume3 was the eroded volume under X ratio = 3.2 from Table 3 of Matsubara et al.[32].

**Table 3 | Comparison of volume estimates based on the same simulated landscape[29].**

| | Hoke et al.[28] method | Reimman-Simpson | PBTH | 'Truth' |
|---|---|---|---|---|
| Volume (m³) | 18,562.00 | 21,716.00 | 21,109.45 | 21,940.71 |
| Overall relative accuracy | 84.60% | 98.98% | 96.21% | 100.00% |

PBTH, progressive black top hat; VN, valley network.
Calculation of the first two columns performed by Brian Hynek and William Nelson, University of Colorado, Boulder, using the same VN area boundary.

function of hydraulic radius, sediment grain size and density, among other things[24]. Although their favoured range of $\alpha$ is 35–60, they pointed out the range of $\alpha$ for large VNs should be 6–35 (ref. 24). Therefore, cumulative water volumes for all assumed values of $\alpha$ considerably exceed the estimate of Rosenberg and Head[24].

## Discussion

Our estimates of the VN volume and water volume are based on the following assumptions: (1) the valley shoulder elevations did not change significantly since their formation; (2) the amount of sediments carried into the valleys from elsewhere (for example, from hillslope by sheet flow) was negligible. In addition, the post-formation infill by other processes such as eolian process or mass wasting were deemed minimum and not considered. Thus, the estimated water volume is the minimum cumulative volume of water required to carve the VNs on Mars globally. Yet, this minimum volume of water is larger than the volume of the hypothesized northern ocean (ranging from 156 to 548 m GEL[10,14]), which suggests that the water must have cycled through the VN system many times (that is, implying an active hydrologic cycle) and the early climate was likely to be wetter than suggested by some previous studies[19,24]. We realize that not all VNs on Mars drain to the northern ocean. If we only include those VNs that drain to the northern lowlands (based on current topography), our estimate of cumulative water needed to carve them would be 3.32 km GEL (based on the topographically derived VNs), still larger than previous cumulative water volume estimates and the volume of hypothesized ocean, suggesting an active hydrologic cycle. As on Earth, the great amount of water recycling needed to carve VNs would probably require a large open water body (ocean) on Mars contemporaneous with the VN formation, and a warm and wet climate to support the active hydrologic cycle. Without an ocean-sized open body of water, it would be hard to imagine the high rate of water cycling suggested by our new estimates. Given the large inventory of water from the perspective of global VNs, and that the age of most of the Martian VNs is more than three billion years old (carved into the ancient Noachian highlands),

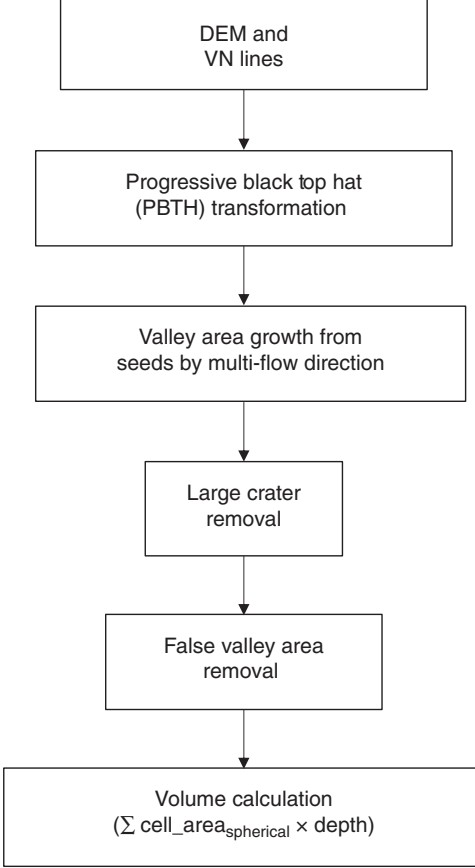

**Figure 1 | Overall flow diagram.** The process of applying the PBTH method to the whole of Mars. It is worth noting that the globe is divided into 20° × 20° tiles and is processed one tile at a time.

the active hydrologic cycle, the warm and wet climate, and the existence of an ocean must have happened early in Martian history.

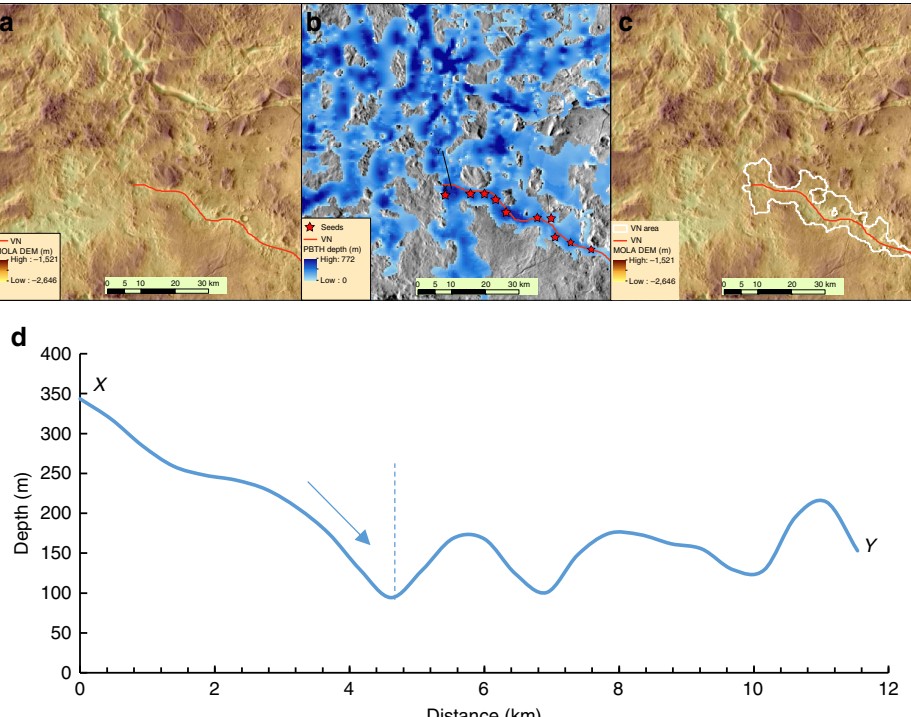

**Figure 2 | Valley area growth from seeds by multi-flow direction. (a)** VN and shaded MOLA DEM (for location, see Fig. 3b). (**b**) VN, thresholded PBTH depth and local maxima 'seeds' for generating VN area following multi-flow direction on depth grid. (**c**) VN, VN area polygon and shaded MOLA DEM. (**d**) Depth profile along XY in **b**. Dashed vertical line indicates where the flow would stop, which defines the boundary of VN area polygon as shown in **c**.

There is no ground truth to assess the real accuracy of our estimation. However, we can establish the confidence of our estimates by comparing them with those of previous studies at the same locations using similar data, for example, Hoke et al.[28] and Matsubara et al.[32] (Table 2). As shown in Table 2, our estimates are generally consistent with these previous studies to within one order of magnitude with the ratio of our estimate to theirs ranging from 0.15 to 1.9. The differences are due to different methodologies used in calculating the volume and how the valley areas were defined. Matsubara et al.[32] used the 75th percentile elevation within a search radius along the valley as the shoulder elevation and minimum elevation within that search window as the valley bottom elevation to estimate the eroded volume. Their estimates are larger than ours, as their method generally results in valley areas larger than ours. If we convert our VN volume estimate of the Evros and Samara, Parana and Loire Valles (first and last rows of Table 2) into volume of sediments, our numbers would be $2.28 \times 10^{12}$ and $9.79 \times 10^{12}$ m$^3$, respectively, much closer to their estimates. Hoke et al.[28] estimated the VN volumes based on manually drawn boundary along the outer walls of the visible valleys and measurements of width and length and estimates of depth. The manually drawn VN boundary is likely different from that derived from PBTH method and may be the primary reason for the differences in volume estimates. Applying the Hoke et al.[28] method to the same simulated landscape with the same VN boundary[29] resulted in a relative accuracy of ~85% (see Table 3, calculation courtesy of Brian Hynek and William Nelson, University of Colorado, Boulder, personal communication). They have recently improved their methodology by using the Reimman sums and the Simpson rule. Again, using the same simulated landscape and same VN boundary, the improved method resulted in a slightly better relative accuracy than ours (by about 2.7%, see Table 3, calculation courtesy of Brian Hynek and William Nelson,

personal communication). However, their new method requires considerably more human intervention and the user still has to provide a valley area boundary, which BPTH method can automatically generate. The small gain in relative accuracy would not make any significant difference in global estimate of the VN volumes.

Thus, we are confident about PBTH method as a robust, accurate and efficient method for estimating the global VN volume. The PBTH method objectively and consistently delineates the VN boundaries and estimates valley depth at pixel level. Because of the automated procedure, we have included all the VNs mapped at the global scale to date. Our results provided an independent source of estimate of global water inventory on Mars. Our result is consistent with a warm and wet early Mars climate and the existence of an ancient northern ocean. If erosion of the VNs required significant chemical or physical weathering to produce transportable sediment, fluvial abrasion of channel beds[33] or transport of appreciable quantities of gravel[34], the required volume of water may have been many times our conservative estimate.

Existing climate models have not been able to reproduce an early Mars climate sufficient to promote an active hydrological cycle[35–37] (see also a recent review by Wordworth[38] and references therein); this has led to hypotheses suggesting accumulation of thick cold-based ice on the equatorial highlands with VN formation during short-lived episodes of top–down melting[39]. If true, this argues against an unfrozen ocean. However, an equivalent amount of erosion and equivalent total runoff would still need to be accounted, which may be challenging to achieve under cold climate scenario[36]. The gap between the geomorphic evidence such as this study, which suggests a warm and wet climate, and the climate models that struggle to get temperature high enough for early Mars[36] still requires further study.

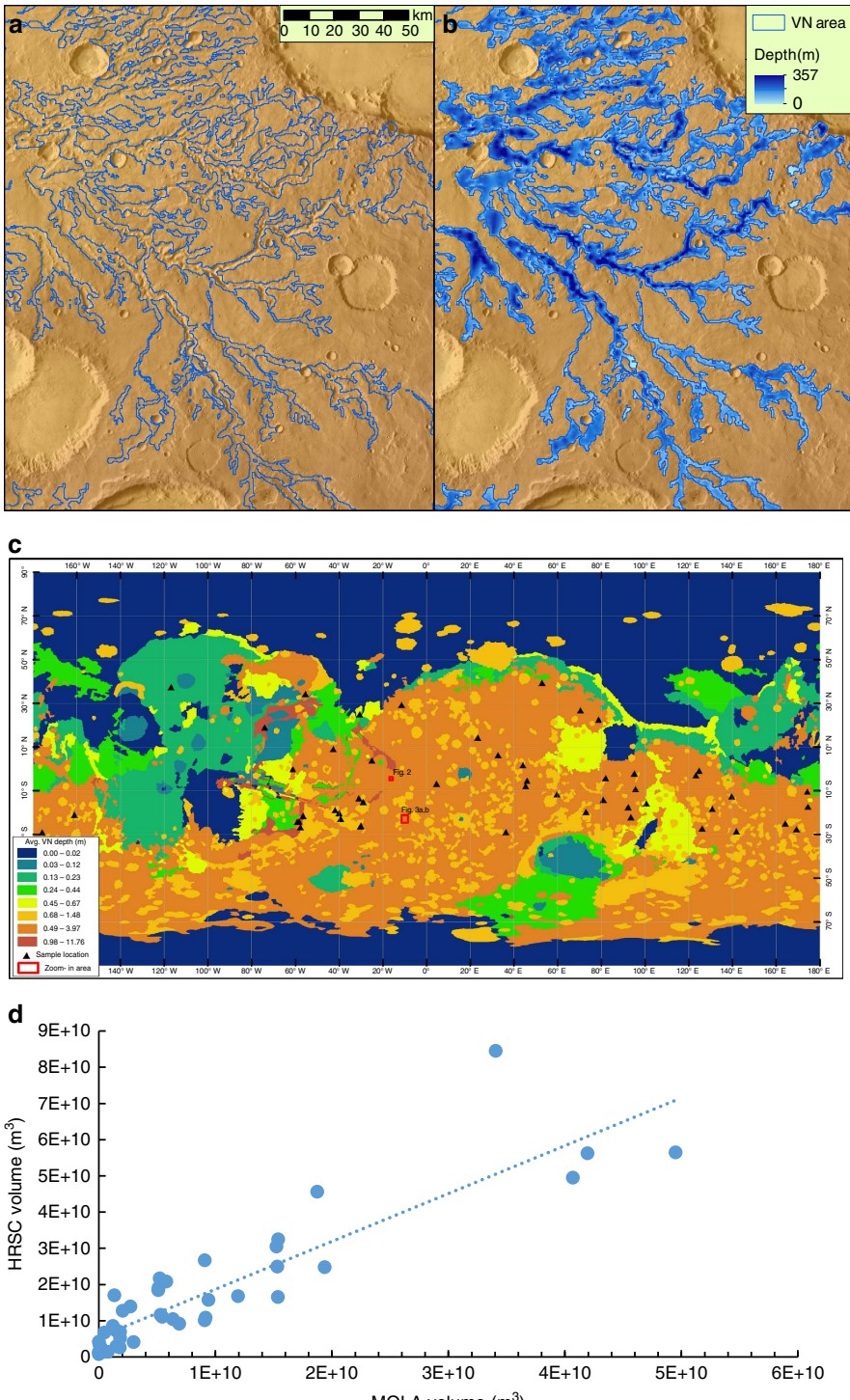

**Figure 3 | Example of PBTH-derived VN depth and spatial distribution. (a)** VN boundary overlain on MOLA DEM shaded relief (see **c** for location). (**b**) VN depth. (**c**) Global distribution of zonal mean VN depth by geologic unit[43]. (**d**) Regression relating volumes extracted from MOLA and HRSC data based on 50 random samples (location shown as triangles in **c**). The regression equation is: $y = 1.3215x + 5 \times 10^9$, $R^2 = 0.79$.

## Methods

**Automation of PBTH method for global Mars application.** We set out to conduct a comprehensive global VN volume estimation through developing a method that is robust, accurate and automated. The PBTH method we adopted integrates techniques used in Lidar data analysis and image processing[29], which allowed us to estimate VN depth at individual pixel level[29,40] and to derive VN volume on a global scale with greater accuracy and efficiency. Our test on simulated landforms has achieved a relative accuracy of 96% and application to Ma'adim Vallis resulted in a volume value higher than previous estimates[29].

The application of the method to the whole of Mars required some special considerations and the process is outlined in the flow diagram in Fig. 1. To make

the processing more tractable, we divided MOLA DEM into 20° × 20° tiles and processed one tile at a time. After all the tiles were processed, they were merged into one global dataset to estimate the global VN volume. To automate the process as much as possible, we took advantage of the previously extracted VN lines. Luo and Stepinksi[16] extracted the global VNs using a morphology-based algorithm, which resulted in VNs that strictly follow topography but may be disconnected at places. Based primarily on images, Hynek et al.[30] manually mapped another version of the global VNs, which are more connected but may not always follow topography. We derived two volume estimates: one based on the VN derived from topography and the other based on the combined VN from the two existing versions using a GIS conflation tool.

**More details of PBTH method.** Black top hat (BTH) transformation is a mathematical morphology transformation used in image processing to extract dark features on an image with varied background[40]. It has been adapted to extract valley depth from DEM data[40] (low elevations of the valleys are the dark features in image terms). Operationally, it involves finding focal maximum of the DEM within a moving circle (opening), then focal minimum of the opening result (closing) and subtracting the original DEM from the closing result[40]. The process essentially creates a pre-incision surface based on the present day topography under the assumption that the elevation of the valley shoulders did not change significantly, which is a reasonable assumption for Mars as the VNs carve into the highland surface[29]. To address the drawbacks of using a single window size in BTH, a series of windows with progressively bigger sizes (from 3 to 11 cells) were used to capture valleys of different sizes more effectively (hence, the name progressive BTH or PBTH)[29]. At the end of each BTH operation, the depth result was thresholded to remove noise. Following algorithms used in LiDAR data processing[29], a slope factor (0.02) was introduced to scale the threshold value according to each progressive window size[29]. The VN depth grids were merged together, taking the maximum value if a cell location had multiple values from different window sizes. This final depth grid represents the depth of all the depressions (essentially an inverse of topography with deeper depression having higher depth value), most of which are indeed VNs, but may also include some non-VN depressions (Fig. 2a,b). To automatically generate the correct VN areas around the VN lines, we started at local maxima of the VN depth grid that are near the VN line ('seeds') and followed a standard multiple flow direction algorithm to grow the area around the 'seeds' to form the VN area polygon (Fig. 2c). Crater depressions that still remain after the multiple flow direction algorithm were removed using location and diameter information from an existing database of craters[41]. Some parts of shallow valleys do not meet the depth threshold, resulting in small gaps in the valley area polygon. These gaps were connected by buffering each pixel of under the VN line with a buffer size that is ten times the depth at that location (that is, assuming the width is about ten times the depth[42], a conservative estimate). More details of the process is documented in the Python source code and is available as Supplemental Material. Finally, visual inspection and manual editing was conducted to remove any false positives not consistent with VN morphology based on Thermal Emission Imaging System images. An example of the final VN area and VN depth is shown in Fig. 3a,b and the zonal mean of VN depth by geologic unit[43] is shown in Fig. 3c, illustrating its global spatial distribution.

**VN volume calculation.** The volume of VNs was calculated by summing the volume of each pixel column (product of depth and area of each pixel) of the valley depth grid inside the VN area polygon. To avoid distortion associated with map projection for the global data, we used the spherical area of each pixel. The DEM data with global coverage is the MOLA DEM at ~463 m per pixel resolution. The High/Super Resolution Stereo Colour Imager (HRSC) DEM data have higher resolution (~75 m per pixel), but with limited coverage. We took 50 random sampled areas where both HRSC and MOLA data are available and extracted VN volumes from both resolutions. A regression line was established between the values at these two resolutions (Fig. 3d) and the final MOLA DEM-derived volume was scaled following the regression line as the estimate of volume under higher HRSC resolution.

**Code availability.** The Python scripts for performing the PBTH volume calculation in ArcGIS can be found in the Supplementary Software. These include readme.txt (Supplementary Software 1), main.py (Supplementary Software 2) and extract_vn_library.py (Supplementary Software 3).

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

## Acknowledgements

This research was supported by NASA Mars Data Analysis Program Grant number NNX13AK65G. We thank V. Baker and two anonymous reviewers for their constructive reviews of manuscript, which greatly improved its quality. We thank B. Hynek and W. Nelson for calculating the volume of valleys in a simulated landscape for comparison.

## Author contributions

W.L. conceived this research study, devised the PBTH algorithm and wrote the paper. X.C. implemented the PBTH and multi-flow direction algorithms in Python, processed the data and contributed to the manuscript preparation. A.D.H. helped with the interpretation of the results and contributed to manuscript preparation.

## Additional information

**Competing interests:** The authors declare no competing financial interests.

**Publisher's note**: 

