## [Peer Review File · Nature Communications]

Reviewer #1 (Remarks to the Author):

The paper considers the entire valley networks from the Ma'adim Vallis region in Mars and proposes a semi-automatic method allowing the comprehensive VN cavity volume estimation.

The method is essentially based on the use of the Black Top Hat transformation, a mathematical tool that has been previously used to estimate VN cavity volume from DEM data. The main innovation here is the implementation of the progressive BTH (PBTH), an algorithm recently published by some of the authors in *Computer and Geosciences*. In order to address the drawbacks of using a single window size in BTH, the PBTH consider a series of windows with progressively bigger sized. This improvement allows capturing valleys of different sizes more effectively. Particularly, the contribution of small VNs to volume estimations, which have not been considered in the volume estimates previously published. The method has been successfully checked on simulated topographies and the estimated volumes have a high accuracy. Several additional aspects allow removing the non-VN depressions and improving the volume estimation: use of standard algorithm to draw watershed basin, consideration of existing crater database and visual inspection and manual editing. The paper also take advantage of the existence of some DEMs of higher resolution and by implementing regression curves from data from different DEM resolution estimate the volume under the higher HRSC resolution. The proposed method is clearly a robust and accurate method allowing the comprehensive VN cavity volume estimation.

Once the VN volume estimated, the water volume required to carve the VNs is estimated based on empirical terrestrial water-to-sediment ratios previously adapted and used in Mars. The estimated water volume is a minimal estimate of water required to carve the VNs and is much larger than the water needed to have a ocean in Mars. From this, the authors suggest a sort of active hydrologic cycle in the past of Mars.

The main novelty of this paper is clearly the application of the PBTH, an algorithm allowing the estimation, for the first time, of the entire VNs volume in Mars. Previous publications presented local estimations based in crude methods. The proposed methodology integrates algorithms used in the LIDAR image treatments which allow to extend the BTH applications. It is of interest not only for future applications on Mars, but also on the Earth, particularly in desert areas. Considering that the used water-to-sediment ratio is valid in Mars, that the valley morphology did not change since its formations and that had not sediments carried into the valleys from elsewhere (VNS are geomorphologic systems isolated from the rest of the landscape), the water volume estimate needed to carve the VNs is a robust and key data. This data has a high value for the live discussion on the existence of water cycles and a past warmer and wetter climate in Mars. The weight of this estimation for such discussion is clearly presented in the paper synthetically and well contextualized in the literature.

Reviewer #2 (Remarks to the Author):

Review of “Ancient Martian Ocean and Wet Climate Supported by New Estimate of Valley Network Volume” Reviewed by Vic Baker, University of Arizona

This paper provides a welcome updating of earlier studies that recognized the importance of extensive hydrological cycling in the origin of valley networks (VNs) on Mars. Using new methodologies the authors have done an excellent job of quantifying the excavated volume of valley networks on Mars and comparing this volume to the volumes of recycled water needed to account for that excavation. There are some minor issues with the writing and clarity on a few points, but overall this is an important contribution that should definitely be published in Nature Communications.

The paper is somewhat confusing in places in regard to the inventory of water on Mars. The 6.08 km GEL noted in line 24 is not an inventory that is appropriate for comparison to the figures listed in line 25. Rather, the 6 GEL is the volume of water recycling ($8.8 \times 10^8 \text{ km}^3$) that was needed to excavate the volume of the VNs ($2.2 \times 10^5 \text{ km}^3$). Dividing the former by the latter yields the inference that the volume of recycled water flushed through to excavate the VNs was about 400 times the volume of the VNs themselves. This is similar to what is observed for fluvial valleys on Earth, like those in the northern part of the island of Hawaii (calculations made by V.C. Gulick). This great amount of recycling suggests that an immense standing body of water on Mars must have been present to generate the global hydrological cycle that is needed for its explanation. In other words, as on Earth, the great amount of water recycling involved in excavating the Mars VNs would seem to require an ocean on Mars contemporaneous with the VN formation. Given that the age of most of the Martian VNs is more than 3 billions years, this was a very ancient ocean.

The list of supporting evidence for the Mars ocean hypothesis (lines 37-39) is incomplete, and the arguments for and against the ocean hypothesis are much more complex than what is presented in lines 37-45. Nevertheless, the short format of this publication justifies a mere acknowledgement that space limitations preclude a more detailed review of the issues.

There are many minor issues of style. I note a few of these below, but I suggest that a more thorough editing be done.

Line 17. Should read “rate of hydrologic cycling” not “cycle”

Line 32. The VNs of Mars are not “river-like.” They are “river-valley-like”

Line 48. Insert a comma after “constrained” and replace “their” with “these”

Line 67. Replace “landform” with “landforms” and insert comma after “96%”

Line 70. Insert “of” after “whole” and insert comma after “considerations”

Lines 75-78. The wording of this sentence is unclear.

Line 119. The VN volume estimate given here ($2.96 \times 10^5 \text{ km}^3$) does not match the figure given in the Abstract ($2.23 \times 10^5 \text{ km}^3$).

Lines 133-136 make the point about the volume being the amount of water recycling.

Lines 145-148. Indeed, yes.

Reviewer #3 (Remarks to the Author):

Luo et al. present a new estimate of the volume of eroded material from valley networks on Mars. Using a fixed water-to-sediment ratio, they translate this valley volume estimate into an estimate for the time-integrated water discharge that flowed through the valleys, which is the main implication of this work. The estimated, time-integrated water volume is equivalent to a 6.08 km global layer, which is roughly 40 times greater than all previous estimates for water volume based on the valley networks.

Based on these claims, this paper would be of interest to readers curious about the climate history of the terrestrial planets, and the geologic history and past habitability of Mars. The general approach adopted in this manuscript was previously used by Carr and Malin [2000], and elaborated

by Rosenberg and Head [2015]. The novel contribution of Luo et al. is the new estimate of global valley volume, derived using a semi-automated image processing approach, the progressive black top hat (PBTH) transformation [Luo et al., 2015]. The results from this mapping approach are shown very nicely in Figure 2. The new valley volume estimate ($2.23 \times 10^5 \text{ km}^3$) is approximately ten times larger than the previous estimate ($2.4 \times 10^4 \text{ km}^3$; Rosenberg and Head [2015]).

The claims of this paper are not fully convincing for two reasons. First, the implication of the title—that the results indicate an ancient Martian ocean—is not supported. The text itself says the inferred water volume is unlikely to have existed all at once, given other lines of evidence (lines 132-133). This paper presents no new evidence that a large body of water existed in the northern hemisphere. Second, the new and dramatically larger valley volume estimate is the core of the paper, and the potential sources of error warrant further quantification.

I will elaborate on the second point and offer some suggestions that could make the valley volume analysis more convincing. The paper argues that the smaller valley volume used by Rosenberg and Head [2015] owes to that study considering only a subset of valley networks. To prove this claim, I think the authors need to show that the new method can reproduce the valley volume estimates for the same subset of valleys used by Rosenberg and Head [2015]. If the method can do this, it would demonstrate that the larger valley volume reported here actually results from considering more valleys. Luo et al. [2015] did compare valley volume estimates for Ma'adim Vallis using the PBTH method to independent estimates by Goldspiel and Squyres [1991] and Gulick [2001]. Although there was a broad agreement in the estimated valley volumes for these three cases, Luo et al. [2015] noted that “both of these previous studies admitted large uncertainties in their estimates.” Importantly, neither Goldspiel and Squyres [1991] nor Gulick [2001] had access to reliable topography data, which first became available from the Mars Orbiter Laser Altimeter dataset. Therefore, in neither their 2015 paper nor the present work have Luo et al. shown how the PBTH method compares with other methods for estimating valley volume using the same topography dataset. This gap makes it difficult to directly compare to the results of Rosenberg and Head [2015]. Lastly, the authors seem to have used a rather liberal definition of a valley, by combining valleys mapped by topography with valleys mapped by images alone (lines 75-79). The latter dataset includes valley traces that do not follow the topography, and may be unreliable for detailed morphologic analysis. It would be useful to know if using only the valley traces mapped from topography, which seem more reliable, yields a valley volume consistent with the value reported here.

In summary, the novelty of the paper, and its significance relative to other related studies, depends heavily on the new valley volume estimate. I recommend further analysis to test the robustness of the valley volume estimate. If this more detailed analysis still suggests a large integrated water discharge compared to previous studies, then the results could represent a significant finding. I believe strongly that the evidence for a Mars ocean is overstated, as this paper does not constrain the amount of water that could have instantaneously pooled in the northern hemisphere.

References

Carr, M. H., and Malin, M. C., 2000, Meter-scale characteristics of martian channels and valleys, *Icarus* 146(2), 366–386.

Goldspiel, J. M., and Squyres, S. W., 1991, Ancient aqueous sedimentation on Mars, *Icarus* 89, 392–410.

Gulick, V. C., 2001, Origin of the valley networks on Mars: a hydrological perspective, *Geomorphology* 37, 241–268.

Luo, W., Pingel, T., Heo, J., Howard, A. & Jung, J. A progressive black top hat transformation algorithm for estimating valley volumes on Mars. *Comput. Geosci.* 75, 17–23 (2015).

Rosenberg, E. N., and Head, III, J. W., 2015, Late Noachian fluvial erosion on Mars: Cumulative water volumes required to carve the valley networks and grain size of bed-sediment. *Planet. Space Sci.* 117, 429–435.

Reviewers' comments:

Reviewer #1 (Remarks to the Author):

The paper considers the entire valley networks from the Ma'adim Vallis region in Mars and proposes a semi-automatic method allowing the comprehensive VN cavity volume estimation.

The method is essentially based on the use of the Black Top Hat transformation, a mathematical tool that has been previously used to estimate VN cavity volume from DEM data. The main innovation here is the implementation of the progressive BTH (PBTH), an algorithm recently published by some of the authors in *Computer and Geosciences*. In order to address the drawbacks of using a single window size in BTH, the PBTH consider a series of windows with progressively bigger sized. This improvement allows capturing valleys of different sizes more effectively. Particularly, the contribution of small VNs to volume estimations, which have not been considered in the volume estimates previously published. The method has been successfully checked on simulated topographies and the estimated volumes have a high accuracy. Several additional aspects allow removing the non-VN depressions and improving the volume estimation: use of standard algorithm to draw watershed basin, consideration of existing crater database and visual inspection and manual editing. The paper also take advantage of the existence of some DEMs of higher resolution and by implementing regression curves from data from different DEM resolution estimate the volume under the higher HRSC resolution. The proposed method is clearly a robust and accurate method allowing the comprehensive VN cavity volume estimation. Once the VN volume estimated, the water volume required to carve the VNs is estimated based on empirical terrestrial water-to-sediment ratios previously adapted and used in Mars. The estimated water volume is a minimal estimate of water required to carve the VNs and is much larger than the water needed to have a ocean in Mars. From this, the authors suggest a sort of active hydrologic cycle in the past of Mars.

The main novelty of this paper is clearly the application of the PBTH, an algorithm allowing the estimation, for the first time, of the entire VNs volume in Mars. Previous publications presented local estimations based in crude methods. The proposed methodology integrates algorithms used in the LIDAR image treatments which allow to extend the BTH applications. It is of interest not only for future applications on Mars, but also on the Earth, particularly in desert areas. Considering that the used water-to-sediment ratio is valid in Mars, that the valley morphology did not change since its formations and that had not sediments carried into the valleys from elsewhere (VNS are geomorphologic systems isolated from the rest of the landscape), the water volume estimate needed to carve the VNs is a robust and key data. This data has a high value for the live discussion on the existence of water cycles and a past warmer and wetter climate in Mars. The weight of this estimation for such discussion is clearly presented in the paper synthetically and well contextualized in the literature.

Thank you!

Reviewer #2 (Remarks to the Author):

Review of "Ancient Martian Ocean and Wet Climate Supported by New Estimate of Valley Network Volume" Reviewed by Vic Baker, University of Arizona

This paper provides a welcome updating of earlier studies that recognized the importance of extensive hydrological cycling in the origin of valley networks (VNs) on Mars. Using new methodologies the authors have done an excellent job of quantifying the excavated volume of valley networks on Mars and comparing this volume to the volumes of recycled water needed to account for that excavation. There are some minor issues with the writing and clarity on a few points, but overall this is an important contribution that should definitely be published in Nature Communications.

The paper is somewhat confusing in places in regard to the inventory of water on Mars. The 6.08 km GEL noted in line 24 is not an inventory that is appropriate for comparison to the figures listed in line 25. Rather, the 6 GEL is the volume of water recycling ($8.8 \times 10^8 \text{ km}^3$) that was needed to excavate the volume of the VNs ($2.2 \times 10^5 \text{ km}^3$). Dividing the former by the latter yields the inference that the volume of recycled water flushed through to excavate the VNs was about 400 times the volume of the VNs themselves. This is similar to what is observed for fluvial valleys on Earth, like those in the northern part of the island of Hawaii (calculations made by V.C. Gulick). This great amount of recycling suggests that an immense standing body of water on Mars must have been present to generate the global hydrological cycle that is needed for its explanation. In other words, as on Earth, the great amount of water recycling involved in excavating the Mars VNs would seem to require an ocean on Mars contemporaneous with the VN formation. Given that the age of most of the Martian VNs is more than 3 billions years, this was a very ancient ocean.

We have modified the text and incorporated some of the language above into the text.

The list of supporting evidence for the Mars ocean hypothesis (lines 37-39) is incomplete, and the arguments for and against the ocean hypothesis are much more complex than what is presented in lines 37-45. Nevertheless, the short format of this publication justifies a mere acknowledgement that space limitations preclude a more detailed review of the issues.

We added some more references.

There are many minor issues of style. I note a few of these below, but I suggest that a more thorough editing be done.

Line 17. Should read “rate of hydrologic cycling” not “cycle”
Changed as suggested.

Line 32. The VNs of Mars are not “river-like.” They are “river-valley-like”
Changed as suggested.

Line 48. Insert a comma after “constrained” and replace “their” with “these”
Changed as suggested.

Line 67. Replace “landform” with “landforms” and insert comma after “96%”
Changed as suggested.

Line 70. Insert “of” after “whole” and insert comma after “considerations”
Changed as suggested.

Lines 75-78. The wording of this sentence is unclear.
Revised to be clear (see lines 169 to 176 in the new version).

Line 119. The VN volume estimate given here ($2.96 \times 10^5 \text{ km}^3$) does not match the figure given in the Abstract ($2.23 \times 10^5 \text{ km}^3$).
Changed and updated.

Lines 133-136 make the point about the volume being the amount of water recycling.

Lines 145-148. Indeed, yes.

Reviewer #3 (Remarks to the Author):

Luo et al. present a new estimate of the volume of eroded material from valley networks on Mars. Using a fixed water-to-sediment ratio, they translate this valley volume estimate into an estimate for the time-integrated water discharge that flowed through the valleys, which is the main implication of this work. The estimated, time-integrated water volume is equivalent to a 6.08 km global layer, which is roughly 40 times greater than all previous estimates for water volume based on the valley networks.

Based on these claims, this paper would be of interest to readers curious about the climate history of the terrestrial planets, and the geologic history and past habitability of Mars. The general approach adopted in this manuscript was previously used by Carr and Malin [2000], and elaborated by Rosenberg and Head [2015]. The novel contribution of Luo et al. is the new estimate of global valley volume, derived using a semi-automated image processing approach, the progressive black top hat (PBTH) transformation [Luo et al., 2015]. The results from this mapping approach are shown very nicely in Figure 2. The new valley volume estimate ($2.23 \times 10^5 \text{ km}^3$) is approximately ten times larger than the previous estimate ($2.4 \times 10^4 \text{ km}^3$; Rosenberg and Head [2015]).

The claims of this paper are not fully convincing for two reasons. First, the implication of the title—that the results indicate an ancient Martian ocean—is not supported. The text itself says the inferred water volume is unlikely to have existed all at once, given other lines of evidence (lines 132-133). This paper presents no new evidence that a large body of water existed in the northern hemisphere. Second, the new and dramatically larger valley volume estimate is the core of the paper, and the potential sources of

error warrant further quantification.

We have changed to title to “New Martian Valley Network Volume Estimate Consistent with Ancient Ocean and Warm and Wet Climate”

I will elaborate on the second point and offer some suggestions that could make the valley volume analysis more convincing. The paper argues that the smaller valley volume used by Rosenberg and Head [2015] owes to that study considering only a subset of valley networks. To prove this claim, I think the authors need to show that the new method can reproduce the valley volume estimates for the same subset of valleys used by Rosenberg and Head [2015]. If the method can do this, it would demonstrate that the larger valley volume reported here actually results from considering more valleys. Luo et al. [2015] did compare valley volume estimates for Ma’adim Vallis using the PBTH method to independent estimates by Goldspiel and Squyres [1991] and Gulick [2001]. Although there was a broad agreement in the estimated valley volumes for these three cases, Luo et al. [2015] noted that “both of these previous studies admitted large uncertainties in their estimates.” Importantly, neither Goldspiel and Squyres [1991] nor Gulick [2001] had access to reliable topography data, which first became available from the Mars Orbiter Laser Altimeter dataset. Therefore, in neither their 2015 paper nor the present work have Luo et al. shown how the PBTH method compares with other methods for estimating valley volume using the same topography dataset. This gap makes it difficult to directly compare to the results of Rosenberg and Head [2015]. Lastly, the authors seem to have used a rather liberal definition of a valley, by combining valleys mapped by topography with valleys mapped by images alone (lines 75-79). The latter dataset includes valley traces that do not follow the topography, and may be unreliable for detailed morphologic analysis. It would be useful to know if using only the valley traces mapped from topography, which seem more reliable, yields a valley volume consistent with the value reported here.

We added the volume based on topographically derived VN, resulting in a minimum global VN volume of $1.74 \times 10^{14} \text{ m}^3$ and minimum cumulative volume of water required of $6.86 \times 10^{17} \text{ m}^3$ (or $\sim 5 \text{ km}$ of global equivalent layer, GEL). These numbers, although slightly smaller than that based on a combined VN, are still the same order of magnitude as before and thus our conclusion remains the same. We also added error estimate based error propagation. See lines 74-89.

We added Table 2 to compare our estimates with those of Hoke et al. (2011) and Matsubara et al. (2015) at the same locations using similar data. The results are generally consistent and the differences are due to different methods and different VN boundaries used.

Applying the Hoke et al. method to the same simulated landform with same VN boundary resulted in a relatively accuracy of $\sim 85\%$ (compare to ours of 96%). Using a new improved method resulted in a slightly better relative accuracy than ours. However, this is based on the VN boundary derived from PBTH method. Our method is automated and the slight increase in relative accuracy using the new method would not make significant change in the global volume estimate. Thus we are confident about our estimates. See page 6-7 starting at line 111.

In summary, the novelty of the paper, and its significance relative to other related studies, depends heavily on the new valley volume estimate. I recommend further analysis to test the robustness of the valley volume estimate. If this more detailed analysis still suggests a large integrated water discharge

compared to previous studies, then the results could represent a significant finding. I believe strongly that the evidence for a Mars ocean is overstated, as this paper does not constrain the amount of water that could have instantaneously pooled in the northern hemisphere.

With the changes outlined above, we believe we have addressed all the concerns raised by the reviewer 3.

References

Carr, M. H., and Malin, M. C., 2000, Meter-scale characteristics of martian channels and valleys, *Icarus* 146(2), 366–386.

Goldspiel, J. M., and Squyres, S. W., 1991, Ancient aqueous sedimentation on Mars, *Icarus* 89, 392–410.

Gulick, V. C., 2001, Origin of the valley networks on Mars: a hydrological perspective, *Geomorphology* 37, 241–268.

Luo, W., Pingel, T., Heo, J., Howard, A. & Jung, J. A progressive black top hat transformation algorithm for estimating valley volumes on Mars. *Comput. Geosci.* 75, 17–23 (2015).

Rosenberg, E. N., and Head, III, J. W., 2015, Late Noachian fluvial erosion on Mars: Cumulative water volumes required to carve the valley networks and grain size of bed-sediment. *Planet. Space Sci.* 117, 429–435.

Reviewer #3 (Remarks to the Author):

The authors have strengthened the manuscript by benchmarking their valley volume estimates against estimates from other sources (Hoke et al., 2011; Matsubara et al., 2015; and Hynek and Nelson). However, it remains somewhat unclear why the water volume estimate here (~ 5 km GEL) is so much larger than the 3 to 100 m GEL from Rosenberg and Head (2015). To be clear, based on the revisions I am satisfied that the authors have demonstrated the accuracy of the method for measuring valley volume. What is uncertain is: (1) How do the valley volume estimates compare to the same valleys measured in the Rosenberg and Head (2015) study? (2) How much of the difference in the water estimate is due to the different approaches to estimating the water-to-sediment ratio? The authors state that there is an order-of-magnitude difference in the valley volume estimates (line 77), but it is unclear that that alone accounts for the water volume difference. These comparisons to the Rosenberg and Head (2015) results don't strike me as particularly laborious. Should the authors choose to make them, I think it would strengthen the manuscript further.

A second point, which I apologize for not noting earlier, relates to the global map of valley network depth (Figure 3a). The map includes large areas that do not drain to the northern lowlands (e.g., Hellas Basin). Therefore, it seems unlikely that water that carved those valleys could have supported a northern ocean. Alternatively, a large body of work suggests the presence of ancient crater lakes (e.g., Goudge et al., 2016). Therefore, I further recommend the authors separate their water volume estimates into categories -- for example, (1) internally drained crater lake basins; (2) other basins that do not drain to the northern lowlands; (3) basins that do drain to the northern lowlands and are more likely to have fed a putative ocean.

As both of these points bear on the main argument of the paper, I recommend further moderate revisions.

Additional minor comments:

Line 75: "(~ 5km GEL)" - This sentence describes the valley volume estimate alone, but "GEL" refers to the water volume estimate that also includes an assumed water-to-sediment ratio. Recommend cutting the water volume estimate here.

Line 148: "Extant." Wrong word; I think you mean "existing."

Line 149: There are many more climate models out there, and further references are needed.

Figures 2 and 3A. Legends and scale bars are small and hard to read.

Figure 2A. The meaning of the light blue arrow and the dashed line are not indicated.

Reviewers' comments:

Reviewer #3 (Remarks to the Author):

The authors have strengthened the manuscript by benchmarking their valley volume estimates against estimates from other sources (Hoke et al., 2011; Matsubara et al., 2015; and Hynek and Nelson). However, it remains somewhat unclear why the water volume estimate here (~ 5 km GEL) is so much larger than the 3 to 100 m GEL from Rosenberg and Head (2015). To be clear, based on the revisions I am satisfied that the authors have demonstrated the accuracy of the method for measuring valley volume. What is uncertain is: (1) How do the valley volume estimates compare to the same valleys measured in the Rosenberg and Head (2015) study?

This is already addressed in Table 2, because Rosenberg and Head's (2015) volume was taken from Hoke et al. (2011) and was compared in Table 2.

(2) How much of the difference in the water estimate is due to the different approaches to estimating the water-to-sediment ratio? The authors state that there is an order-of-magnitude difference in the valley volume estimates (line 77), but it is unclear that that alone accounts for the water volume difference. These comparisons to the Rosenberg and Head (2015) results don't strike me as particularly laborious. Should the authors choose to make them, I think it would strengthen the manuscript further.

We added a paragraph at the end of Result section (lines 92-99). Using Rosenberg and Head's (2015) empirical method (their Eq. (4)) and our most conservative VN volume estimate, we arrived at cumulative water volume estimates ranging from 0.6 km to 11 km GEL, depending on alpha value (60 to 6, respectively). So different methods produced consistently larger cumulative water volume than previous estimates. We don't have access to their raw data to produce a histogram to show the scatter of the values, but our estimate is right in the middle of the possible range. We prefer our simple approach.

A second point, which I apologize for not noting earlier, relates to the global map of valley network depth (Figure 3a). The map includes large areas that do not drain to the northern lowlands (e.g., Hellas Basin). Therefore, it seems unlikely that water that carved those valleys could have supported a northern ocean. Alternatively, a large body of work suggests the presence of ancient crater lakes (e.g., Goudge et al., 2016). Therefore, I further recommend the authors separate their water volume estimates into categories -- for example, (1) internally drained crater lake basins; (2) other basins that do not drain to the northern lowlands; (3) basins that do drain to the northern lowlands and are more likely to have fed a putative ocean.

We estimated the cumulative water volume based on the topographically derived VNs draining to the northern lowlands and the result is ~ 3 km, still larger than previous estimate of cumulative water volume running through the VNs and the volume of the hypothesized ocean. (see line 111-115)

As both of these points bear on the main argument of the paper, I recommend further moderate revisions.

Additional minor comments:

Line 75: “(~ 5km GEL)” - This sentence describes the valley volume estimate alone, but “GEL” refers to the water volume estimate that also includes an assumed water-to-sediment ratio. Recommend cutting the water volume estimate here.

Done.

Line 148: “Extant.” Wrong word; I think you mean “existing.”

Changed to “existing”.

Line 149: There are many more climate models out there, and further references are needed.

We added the Forget (2013) paper and refer to the recent review paper of Wordworth (2016) and references therein.

Figures 2 and 3A. Legends and scale bars are small and hard to read.

Changed as suggested.

Figure 2A. The meaning of the light blue arrow and the dashed line are not indicated.

Blue arrow was accidentally left there and is now removed. The meaning dashed line was already explained in the figure caption.

In addition to the above changes, we also changed labels for the “volume” columns in Table 1 slightly to “volume or mass” as there is a row that contains mass and we removed the GEL number for that row. We also added the Baker et al. (2015) review paper to the reference.

REVIEWERS' COMMENTS:

Reviewer #3 (Remarks to the Author):

I am satisfied that the points raised in the last review have been reasonably addressed.